psychology/cognition/behaviour

gaze following, attachment, maternal postpartum depression, longitudinal, social context, infant

**Author for correspondence:**
Kim Astor
e-mail: kim.astor@psyk.uu.se

# Social and emotional contexts predict the development of gaze following in early infancy

Kim Astor[1], Marcus Lindskog[1], Linda Forssman[1], Ben Kenward[2], Mari Fransson[1], Alkistis Skalkidou[3], Anne Tharner[4], Juliëtte Cassé[5] and Gustaf Gredebäck[1]

[1]Uppsala Child and Baby Lab, Department of Psychology, Uppsala University, Uppsala, Sweden
[2]Department of Psychology, Oxford Brookes University, Oxford, UK
[3]Department of Women's and Children's Health, Uppsala University, Uppsala, Sweden
[4]Department of Clinical Child and Family Studies, Vrije Universiteit Amsterdam, Amsterdam, The Netherlands
[5]Independent Researcher, Amsterdam, The Netherlands

  KA, 0000-0001-8053-8465

The development of gaze following begins in early infancy and its developmental foundation has been under heavy debate. Using a longitudinal design ($N = 118$), we demonstrate that attachment quality predicts individual differences in the onset of gaze following, at six months of age, and that maternal postpartum depression predicts later gaze following, at 10 months. In addition, we report longitudinal stability in gaze following from 6 to 10 months. A full path model (using attachment, maternal depression and gaze following at six months) accounted for 21% of variance in gaze following at 10 months. These results suggest an *experience-dependent* development of gaze following, driven by the infant's own motivation to interact and engage with others (the *social-first* perspective).

## 1. Introduction

Gaze following, the ability to synchronize visual attention with others based on their gaze direction [1,2], is an ability with a broad phylogenetic base, present in a wide range of species such as non-human primates [3] and canines [4]. In humans, this ability has been associated with social learning in infancy (e.g. language [5–8]) and emotional regulation in toddlers [9]. From a clinical perspective, diminished gaze following is a defining feature of autism [2,10].

To date, most empirical studies that target the ontogenetic development of gaze following have focused on generic development, and few studies have targeted the environmental context within which gaze following emerges. The ability to follow gaze starts to emerge at three months of age [11–13] and becomes more prominent from six months onward [13–15]. During the first year after birth, gaze following improves in many aspects. For example, the latency to follow others' gaze is reduced [16] and infants narrow down to focus more on eyes as the most important marker of others' overt attention (as opposed to movements of the head [17]). With respect to individual differences, Morales *et al.* [7] reported partial stability in gaze following from 8 to 18 months, whereas Gredebäck *et al.* [13] demonstrated clear effects on a group level but found no individual stability in gaze following from two to eight months.

From a theoretical standpoint, a distinction can be made between two broad frameworks, the *experience-expectant* and *experience-dependent*, with different views on how gaze following emerges. According to the *experience-expectant* framework, infants are pre-wired to follow gaze, suggesting that gaze following emerges either independent of, or with a minimum of visual experience (as a pre-potent response [18]). Examples of theories within this framework are those suggesting that domain-specific neural encoding modules allow infants to detect the direction of others' gaze [19]. Others suggest that gaze following is the result of an evolutionary pressure to learn certain types of information from others and that infants possess a tightly genetically canalized understanding of communicative referential signals and tend to follow gaze when preceded by ostensive signals [20]. Furthermore, there are theories suggesting that early gaze following is based on domain-general low-level perceptual cueing, that just happens to lend itself well to gaze following [21], and that more sophisticated mechanisms later override the rudimentary gaze cueing as the child matures cognitively [22]. From an *experience-expectant* viewpoint, the initial development of gaze following is independent of infants' environment and should therefore be resistant to suboptimal social or learning contexts, given a minimum amount of social interactions. Therefore, we should observe the same levels of gaze following across typically developing infants that are treated well by well-functioning parents, that is, in a typical sample of infants across cultures. Qualitative and quantitative differences in interactions—such as those related to depressed parents at risk of reduced social interactivity [23], or relationships characterized by non-secure attachment (associated with less maternal sensitivity, synchrony and tendencies to explore [24,25])—should have no significant impact on the development of gaze following. Tentative support for an *experience-expectant* framework can be found in studies demonstrating face preferences in third trimester fetuses [26] and newborn infants [27], and studies demonstrating the presence of gaze cueing in newborns (responding faster to a target appearing in the direction cued by gaze than in the opposite direction [28]). It has also recently been demonstrated that gaze following develops in a normative manner in contexts where infants have fewer opportunities to engage in gaze following [29,30], findings that can be interpreted as supporting the *experience-expectant* framework (however, see Discussion for alternative interpretations).

In contrast, the *experience-dependent* framework suggests that the development of gaze following is dependent on the quality and/or quantity of infants' experiences and that it develops through interaction with others. In line with this notion, it has been demonstrated that preterm infants follow gaze to a degree that can be expected from their amount of social experience, not postmenstrual age [31]. One potential mechanism driving this development is *reinforcement learning*. According to the *reinforcement* theory, gaze following is the result of domain-general learning processes where following the gaze of someone may lead to the occasional reward. People often look at interesting events and, as such, following their gaze can be rewarding and thus may promote further gaze following [32–37]. Other theories within the *experience-dependent* framework suggest a *social-first* mechanism, according to which infants follow gaze because of a motivation to interact and interpret others behaviour [13,38,39] combined with enhanced attention to faces and social interaction partners [40]. Infants' tendencies to follow gaze is argued to be dependent on internal motivation to engage with others, see their goals, and understand their actions (and the perceptual ability to detect and extrapolate gaze direction). *Experience-dependent* theories suggest that negative external factors that may be related to the infant's early caregiving environment, such as the occurrence of maternal postpartum depression and insecure infant–mother attachment, might have a negative impact on infants' social motivation or reward expectations, which in turn, diminishes the tendency to follow gaze. Therefore, lower degrees of gaze following are expected in infants who are insecurely attached and/or have mothers with elevated postpartum depression symptoms. In support of this notion it has been demonstrated that infants of depressed mothers show reduced gaze activity during face-to-face interaction [41]; however, no prior studies have specifically examined the relation between gaze following and the quality of the social context of the child, relating gaze following to phenomena such as maternal depression, or attachment during infancy.

In this study we will contrast two theoretical frameworks and assess the degree to which the development of gaze following is based on an *experience-dependent* or an *experience-expectant* process. We used the longitudinal dataset from the BASICchild project [42]. Other than gaze following, this dataset includes measures of maternal postpartum depression and infant–mother attachment. We have selected these measures because information about their impact on gaze following is missing, and because they can be viewed as proxies for infants' social and emotional environment, as mothers generally represent the most significant social interaction partner early in life. Before moving to the concrete details of this study we will take a brief look at these measures.

Maternal postpartum depression is defined in the DSM-5 as a major depressive episode with onset within four weeks postpartum [10]. Within clinical practice and research, however, the onset time frame typically covers the first year postpartum [43]. The prevalence varies with setting and definition but is estimated to affect 6–19% of mothers [44,45]. The condition has been linked to significant negative consequences for the woman herself and also for her child, including lack of maternal emotional availability, inadequate maternal responsivity and attachment difficulties [46]. Depressed mothers have been found to spend significantly less time in face-to-face interactions with their child and during interactions they display more negative affect compared to non-depressed mothers [47,48]. Elevated levels of maternal postpartum depression may have detrimental effects on mother–infant relationships and attachment [49].

Turning to our next measure, attachment: a secure attachment relationship enables the child to use carers as a buffer against stress and as a secure base from which to explore [24,50]. The process of infant attachment formation begins already at birth [50] and infants start to develop stable attachment strategies at around six months of age [51,52], though attachment quality is often measured at 12 months of age with the strange-situation procedure (SSP [24]), a time point when most children have developed sufficient motor skills to facilitate standardized observation of central attachment behaviours. Attachment bonds can be categorized as *secure*, *insecure-avoidant*, *insecure-resistant* [24], or as *disorganized* (i.e. lack of a coherent attachment strategy [53]). Approximately 56–80% of infants are categorized as securely attached [25,54]. Secure attachment status in infancy has been related to increased sensitivity to emotional facial expressions [55,56] and later in childhood it has also been related to better emotional understanding (for a review, see [57]). In the attachment literature, infant–mother attachment is believed to be primarily driven by the mother's response when the attachment system in the infant is activated, as in SSP [24,58]. Thus, attachment style can be seen as an indirect measure of the quality of infant's social and emotional environment. However, so far, little research has investigated how the development of fundamental social skills, such as gaze following, is linked to individual variation in infant–parent attachment quality.

We have outlined how two distinct theoretical frameworks make different predictions about the early development of gaze following. However, the lack of studies targeting infants' environmental contexts has made it challenging to disentangle them. The current study aims to (i) describe the development of gaze following in relation to the infant's own social and emotional environment, in this case, variation in maternal postpartum depression symptoms and infant–mother attachment quality, and in extension (ii) assess whether the onset and future development of gaze following is best described as an *experience-expectant* or an *experience-dependent* process.

# 2. Methods

## 2.1. Participants

The sample consisted of all participants from the BASICchild cohort, a longitudinal project that followed 118 infants from 6 to 30 months of age, collected in Uppsala, Sweden from 2014 to 2018 (for a project description, see [42]; for details about the sample, see electronic supplementary material).

Families in the BASICchild cohort visited the Uppsala Child and Baby Lab when their infant was 6 months (N = 118, M age = 185 days, s.d. = 7 days, range 170–203 days), 10 months (N = 110, M age = 302 days, s.d. = 9 days, range 289–326 days) and 12 months (N = 112, M age = 368 days, s.d. = 18 days, range 320–435 days); additional data collection was performed at 18 and 30 months, but these time points do not include any of the measures used in this study. A laboratory session ranged from 1.5 to 4 h (different for each time point) and included eye-tracking tasks, structured observation tasks (i.e. infant and experimenter/parent interactions), parent–child free play and breaks. The parents also filled out a series of questionnaires at each time point.

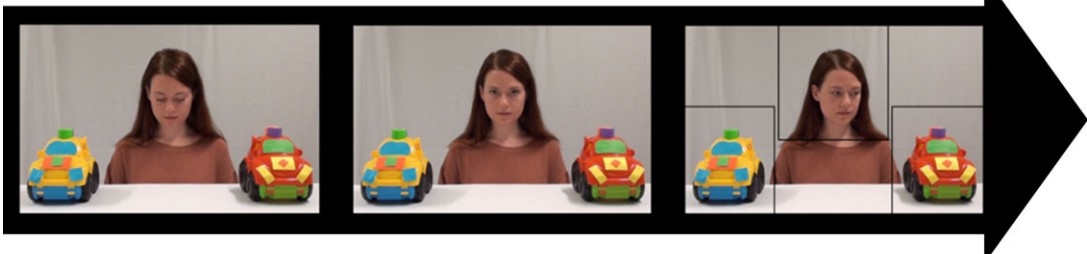

**Figure 1.** One of the three actors who contributed to the stimuli in the gaze following task. First, the actor looked down (left image), then raised her head to look at the infant (middle image) before she turned her head to look at either one of the two objects, left or right (right image). The three black squares within this image represent the AOIs. The black arrow in the background indicates the general time flow of the trial.

The study was conducted in accordance with the 1964 Declaration of Helsinki ethical standards and approved by the local ethics committee. All parents were required to provide written and verbal consent before each visit. After each visit, parents were provided a gift voucher worth approximately €30 as compensation.

## 2.2. Measures and procedures

### 2.2.1. Gaze following

The gaze following task [6,8,11,30,59] consisted of six trials. Each trial was preceded by a central fixation grabber (to attract the infants' gaze to the centre of the screen). On each trial, the infants were presented with a video of a female actor sitting behind a table. Two toy objects were placed to the left and the right of the actor. The actor was facing down during the initial phase of the video (2 s). After a 'beep' signal (0.7 s) the actor raised her head and looked straight into the camera (simulating eye contact with the infant; 2 s). After this phase, she turned her head approximately 45° and gazed to one of the targets (less than 1 s) and kept her gaze at the toy (5 s). The direction of the actors' gaze, left (L) and right (R), was presented in a fixed counterbalanced order across trials (LRRLRL or RLLRLR). A Tobii TX300 (60 Hz) eye-tracker was used to measure infants' gaze in a bright room adapted to limit visual distractions. Infants were seated on their parent's lap at an approximate distance of 60 cm from the monitor. Before the stimuli started a 5-point calibration [60] was performed.

### 2.2.2. Gaze following data analysis

Gaze data was exported from Tobii Studio and processed in Matlab version R2017b (9.3.0.713579) using TimeStudio (version 3.18 timestudioproject.com [61]). All analyses can be downloaded via provided links, found in the Data accessibility section, and run in the TimeStudio environment. Rectangular areas of interest (AOI) were created around the actor and the targets (figure 1). Infants' first head-to-target gaze shift was measured from the onset of the actor's head turn to the end of the trial. Only trials where infants made a head-to-target gaze shift were considered valid. A gaze following difference-score was calculated by subtracting the number of incongruent gaze shifts from the number of congruent gaze shifts [32,59,62–64]. This method here provides a score that ranges from a minimum of −6 to a maximum of 6. Participants who contributed at least one valid trial were included in the analyses.

We acknowledge that some studies use other measures (in addition to the first gaze shift) to assess gaze following. Hernik & Broesch [29], for example, also included frequency of looks and looking duration at the cued object. However, the first gaze shift, the actual reorientation of attention, is the principal measure of gaze following [15] most often used in the literature [12,21,32,59,62].

On a similar note, some studies use a proportion score instead of a difference score. Both measures have strengths and limitations and no measure captures individual differences better than the other (they often converge). In broad strokes, they differ in that a difference score requires participants to provide data on all trials to get a 'perfect' score. In contrast, a proportion score allows participants with a less reliable score (fewer number of valid trials) to heavily impact the analysis, as the likelihood of a

'perfect' score (i.e. 100% of trials pointing in the same direction) increases as the number of valid trials decrease. To avoid this problem a strict inclusion criterion is often used together with a proportion score; Senju & Csibra [15] for example used three trials. When testing an infant at a single time point, a stricter inclusion criterion works well. However, the longitudinal project from which the data is drawn selected the difference score together with one trial minimum for inclusion in order to prioritize the integrity of the data matrix, diminishing the number of missing data.

### 2.2.3. Social context measures

Maternal postpartum depression was assessed using the Edinburgh Postnatal Depression Scale (EPDS [65]) at 6 weeks, 6 months and 12 months after child delivery. The SSP [24] was used to assess infant–mother attachment quality when the child was 12 months old. Two certified coders rated attachment according to the ABCD classification, providing a B (secure) versus ACD (non-secure) score. For more information about the social context measures, including data reduction, see the electronic supplementary material.

## 2.3. Statistical analysis

Below, we first report descriptive statistics and zero-order correlations. Next, we evaluated all six measures (gaze following at 6 and 10 months, maternal postpartum depression at 6 weeks, 6 months and 12 months, and attachment measured at 12 months) in a full path model, allowing all concurrent and prior variables to impact all other concurrent and future variables. Attachment was seen as a stable dimension (see introduction) and was allowed to impact all other variables. Model fit was evaluated based on five parameters. Values in square brackets indicate conventional criteria for good model fit [66,67]: A $\chi^2$-test assessed whether there was a difference between data produced by the model and the actual data (non-significant result indicates good fit). Further, we used the comparative fit index (CFI [>0.95]), Tucker Lewis index (TLI [>0.95]), root mean square error of approximation (RMSEA [<0.06]) and standardized root mean square residual (SRMR [<0.08]). To account for missing data, the model was fitted using full information maximum likelihood. We used a generalized Cook's distance to identify multivariate outliers for the full model, and cases with a Cook's distance greater than 1 were removed. This resulted in the exclusion of four cases. The model was fitted with the *lavaan* package (version 0.6-3, [68]) in R (version 3.4.1, [69]) in the RStudio environment (version 1.0.153, [70]).

### 2.3.1. Data management strategy

Before conducting any analyses within the BASICchild project, the proposal must undergo an internal pre-registration process. This process concerns which variables to assess, which analyses we plan to conduct, how data will be handled and which dependent variables will be used. This procedure was set in place prior to the proliferation of pre-registration platforms available today. Initially, we were interested in gaze following and its relation to maternal postpartum depression. After conducting initial correlations, and having them discussed within the BASICchild group, it was agreed (given the association between postpartum depression and infant–mother attachment [71] to include attachment as there was a concern that it would be captured indirectly via the postpartum depression score. No other variables were analysed in this study other than those reported here, to avoid the false-discovery rate inflation associated with mass-correlations. Two other studies have been published from the BASICchild project, focusing on action prediction, action evaluation and motor development [42] and on the relation between action prediction, action evaluation in infancy and later executive control [72] using other variables than those reported here.

# 3. Results

## 3.1. Descriptive statistics and zero-order correlations

Descriptive statistics and zero-order correlations are summarized in table 1. These initial analyses revealed a significant positive longitudinal relationship (Pearson correlation [r]) between gaze following at 6 and 10 months ($r = 0.20$, $p = 0.04$). Using Spearman correlation ($r_s$), we found that attachment classification (B vs ACD) was significantly correlated with gaze following at six

**Table 1.** Zero-order correlations between variables, means, standard deviations and N.

| | 1 | 2 | 3 | 4 | 5 | 6 |
|---|---|---|---|---|---|---|
| 1. attachment[a] | — | | | | | |
| 2. postpartum depression—6 weeks[a] | −0.13 | — | | | | |
| 3. postpartum depression—6 months[a] | −0.01 | 0.56*** | — | | | |
| 4. postpartum depression—12 months[a] | −0.06 | 0.59*** | 0.60*** | — | | |
| 5. gaze following 6 months | 0.21* | −0.02 | 0.03 | 0.02 | — | |
| 6. gaze following 10 months | 0.18† | −0.10 | −0.19† | −0.31** | 0.20* | — |
| mean | 0.56 | 1.81 | 1.67 | 1.57 | 0.56 | 1.28 |
| standard deviation | 0.50 | 0.58 | 0.50 | 0.52 | 2.01 | 2.03 |
| N | 112 | 117 | 109 | 93 | 110 | 107 |

***$p < 0.001$, **$p < 0.01$, *$p < 0.05$, †$p < 0.1$.

Note: N varied between correlations within a range of 87–111. Measures of maternal postpartum depression had a skewed distribution and attachment was classified on a dichotomous scale (marked [a]). All correlations where at least one of the variables is marked with an [a] were conducted using Spearman's *rho* instead of Pearson's *r*.

months ($r_s = 0.21$, $p = 0.03$). Having a secure (B) attachment was associated with more gaze following then having a non-secure (ACD) attachment classification. This relation did not reach significance in relation to gaze following at 10 months ($r_s = 0.18$, $p = 0.07$). Further, elevated levels of maternal postpartum depression at 12 months were significantly negatively correlated with gaze following at 10 months ($r_s = −0.31$, $p < 0.01$) but the relation between maternal postpartum depression at six months and gaze following at 10 months did not reach significance ($r_s = −0.19$, $p = 0.06$). Finally, there were significant positive longitudinal relationships between all measures of maternal postpartum depression. All other correlations yielded *p*-values > 0.1 and are only reported in table 1. To control for age at the 12-months visit, ranging from 320 to 435 days, partial correlations between gaze following at 10 months to attachment were conducted ($r_s = 0.21$, $p = 0.04$), and maternal postpartum depression at 12 months to gaze following at 10 months ($r_s = −0.32$, $p < 0.01$), both showing that age cannot explain the relations.

Single-sample *t*-tests against chance level performance (0) revealed that infants followed gaze at both visits (6 months: $M = 0.56$, s.d. $= 1.00$, $t_{110} = 2.95$, $p = 0.004$; and 10 months: $M = 1.29$, s.d. $= 2.04$, $t_{105} = 6.53$, $p < 0.001$). A paired *t*-test further revealed a significant increase in gaze following between 6 month and 10 months ($t_{98} = 2.96$, $p = 0.004$).

## 3.2. Path model

The model, illustrated in figure 2, had a good model-to-data fit, as indicated by a non-significant $\chi^2$-test ($p = 0.27$) and four indices of fit reaching criteria that are considered good model fit (CFI = 0.995 [>0.95], TLI = 0.960 [>0.95], RMSEA = 0.052 [<0.06], SRMR = 0.021 [<0.08], values in square brackets indicate conventional criteria for good model fit, see Statistical analysis).

There are four significant paths in the model: postpartum depression at 12 months to gaze following at 10 months ($p = 0.02$); attachment to gaze following at six months ($p = 0.03$); postpartum depression at six weeks to postpartum depression at six months ($p < 0.001$); and postpartum depression at six months to postpartum depression at 12 months ($p < 0.001$). This model as a whole explained 21% of the variance in gaze following at 10 months and 6% in gaze following at six months.[1] The script for the analysis is available in the electronic supplementary material. A table with all path coefficients together with their associated *p*-values and standard errors can be found in the electronic supplementary material.

[1]The main findings from the model are replicated when using a proportion score together with a stricter inclusion criterion (three trials) in accordance with Senju & Csibra [15], see Gaze following data analysis. There are significant paths between our measures of postpartum depression, between attachment and gaze following at six months, and between postpartum depression at 12 months and gaze following at 10 months. The effect sizes and explained variance are essentially unaffected and this model also met the five criteria we use to assess model fit.

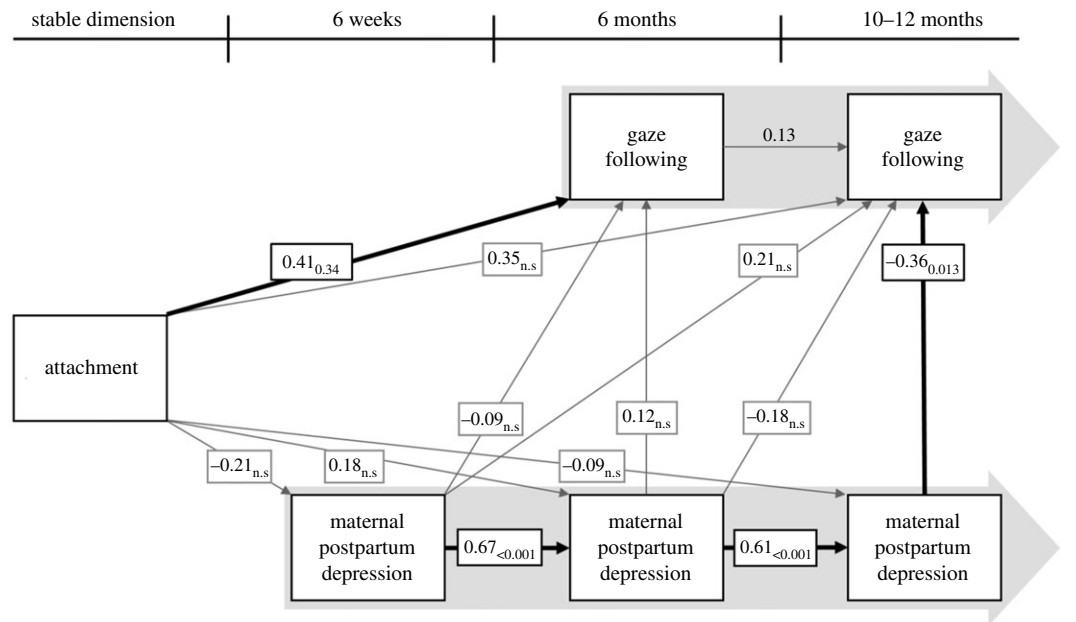

**Figure 2.** Illustration of the path model, structured as we conceptualize our measures to be related. Solid black arrows represent significant paths. Grey thin arrows are non-significant paths. The numbers accompanying each arrow are the standardized path coefficient (left) and its associated *p*-value (right). The wide light grey arrows in the background suggest a timeline, separating the different longitudinal measures.

## 4. Discussion

We demonstrate that secure attachment and lower levels of maternal postpartum depression are associated with more gaze following. More specifically, emerging gaze following, at six months, is more frequent in infants classified with a secure infant–mother attachment relationship compared to those from an insecure and/or disorganized dyad. Later gaze following, at 10 months, is more frequent in infants whose mothers have low levels of postpartum depression. In addition, zero-order correlations demonstrate longitudinal stability in gaze following from 6 to 10 months.

From a theoretical standpoint, we made a distinction between two broad frameworks with different views on how gaze following emerges. On the one hand, the *experience-expectant* framework suggests that infants' ability to follow gaze is tightly genetically canalized (e.g. [19–21]). According to this framework, gaze following is a basic social or perceptual skill with deep phylogenetic roots available to the vast majority of infants and not requiring extensive high-quality social interactions or optimal learning contexts to emerge. In the current study we demonstrate clear associations between infants' social context and the onset and further development of gaze following, a finding at odds with the *experience-expectant* framework.

On the other hand, the *experience-dependent* framework suggests that experience is critical to early developing gaze following [32,40], a claim supported by our findings. Within the *experience-dependent* framework there are two main lines of theoretical perspectives: *reinforcement learning* [32–37] and *social-first* [13,38–40]. The main difference between these perspectives relates to whether they regard gaze following as a learned skill that is motivated by external rewards or a manifestation of infants' growing motivation to engage with others.

The current study alone cannot disentangle the two *experience-dependent* perspectives. It seems plausible that maternal depression and the quality of the mother–infant attachment relationship provide a foundation for, and heavily impact, the social environment of the child, particularly as mothers (at least in the Swedish context where the study is conducted) are the primary carers for most of the infant's first year after birth [73]. It is possible that the social environment impacts the motivation to engage with others and follow their gaze (according to the social-first perspective) or that high-quality social interactions provide more learning opportunities due to stringent and clear associations between the mother's gaze and rewarding external objects or events. In an attempt to separate these perspectives, we rely on two recent studies by Senju *et al.* [30] and Hernik & Broesch [29]. In a UK sample, Senju *et al.* demonstrate that gaze following develops in a similar way whether the

infants' parents are blind or not. Infants of blind parents follow gaze to the same degree as infants of seeing parents. The study does not find any indication that emerging gaze following is driven by the amount of experience infants have had with seeing adults. It should be noted that Senju *et al.* [30] used two measures to assess gaze following. They found no effect of having a blind parent on the principal measure of gaze following, the gaze shift, and though they did find differences in looking time at the attended object, this was only after the infant's first birthday, not at the earlier time point. In a similar manner, Hernik & Broesch [29] demonstrated that infants in Vanuatu, where face-to-face interactions are less common, follow gaze to a similar degree as infants in a western context.

These two studies complement the current one. The results [29,30] suggest that *experience-dependent reinforcement learning* does not drive gaze following, as long-term differences in experience (between families with seeing or blind parents, or between cultures that vary in the amount of face-to-face interactions) do not seem to impact the frequency of gaze following (cf. [31]). Instead, these studies can be interpreted as support for an *experience-expectant* process (where gaze following emerges independent of experience) or an *experience-dependent* process driven by social motivation (not driven by the frequency of exposure but an interest in other people, driven by the quality of prior social interactions). However, learning does not necessarily have to be conceptualized as a slow process. In fact, some studies suggest that infants may acquire gaze following through a rapid learning process (e.g. [32]) where only a few exposures of stimulus–response events leading to positive reinforcement (gaze direction, looking in the direction of gaze, interesting sight) are needed for infants to learn gaze following. This view on *reinforcement learning* (implying a pre-potent stimulus response) instead suggests an *experience-expectant* process. If this is the case, gaze following should be less vulnerable to suboptimal social learning contexts, such as blind parents. This is where the studies converge. A slow *experience-dependent* reinforcement process is less likely given previous research [29,30] showing that amount of experience does not affect amount of gaze following. At the same time, the results from the current study are not consistent with an *experience-expectant* process (such as rapid reinforcement) as it suggests that qualitative differences (within normal variation) in social and emotional environment impact the degree to which infants engage in joint attention through gaze following. Together the studies favour the idea that gaze following is driven by infants' social motivation to interact with other people, an *experience-dependent* process grounded in the quality of the social relation between an infant and her mother (parents).

Irrespective of theory, perhaps the most important finding from the current study is that infants' tendency to follow other peoples' gaze is impacted by maternal depression and the quality of the infant–mother attachment. As gaze following provides a bridge to learning about the world, a diminished gaze following ability or tendency points to a mechanistic explanation for what has been long known, that infants with depressed mothers and children with insecure attachment relationships face profound and potentially lifelong challenges [74,75]. These findings highlight the importance of the parent–infant interaction for children's social development and illustrate the challenges faced by infants living with depressed parents and in families where social support and attachment relationships are less than ideal. Fortunately, this tentative developmental path can be changed by training parental behaviour as a way to avoid negative effects in children and providing support for parents who are unable to provide adequate social support for their child [76,77].

It is important to point out that these effects of social context were observed in otherwise well-functioning families living in a Swedish university town. Postpartum depression is a phenomenon that is relatively evenly distributed across the population (see electronic supplementary material) and our sample has prevalence comparable to the typical population. In other populations the instances of general depression might be substantially higher and it is also possible that particularly vulnerable families (with low socioeconomic status (SES), experience of war or migration, families with abuse or domestic violence) will experience higher rates of depression [78] and insecure attachment [79]. It is possible that the effects observed here would be elevated in some of these families. More work that assesses the impact of social context on gaze following in more diverse settings is needed. Contrary to our concern, we found no support for our notion that attachment was captured indirectly by maternal postpartum depression and we find no association between the measures in the current study (cf. [71], though this association is not always found, especially in otherwise low-risk samples [80]). Differences in assessment age might be one factor explaining why some studies report an association while others do not [71]. Interestingly, both maternal depression and attachment contribute to explain gaze following, indicating that these measures in fact tap into different aspects of infants' social and emotional environment, and further, that they seem to contribute uniquely at different time points. Postpartum depression seems to affect gaze following at 10 months exclusively. Attachment, on the

other hand, only predicts gaze following with statistical significance at six months, not at 10 months, though the statistical difference between the predictions is small. In summary, our results show that the quality of dyadic infant–mother relations is associated with infants' early social development. We demonstrate that securely attached infants tend to follow more gaze at six months and that postpartum depression is associated with less gaze following at 10 months. From a practical standpoint, the results depict some specific challenges that infants in more impoverished social contexts may face and suggest how it may affect their development. In light of Senju et al. [30] and Hernik & Broesch [29], we believe that this study has some theoretical merit: though none of these studies provides conclusive theoretical evidence in isolation, we interpret all three studies to converge on support for an experience-dependent social-first theory of gaze following. By extension, these studies together pose a serious challenge to theories suggesting that gaze following is the result of an experience-expectant process or reinforcement learning. We conclude that gaze following seems to emerge from an experience-dependent process, and more specifically, from infants' own social motivation.

Ethics. The study was approved by the local ethics review committee (EPN) in Uppsala, Sweden (permit number 2013/423), and conducted in compliance with the 1964 Helsinki Declaration. Participating families provided written consent (from all legal parents) prior to the start of the study and at each subsequent visit. The study did not use animal subjects or tissues.

Data accessibility. The eye-tracking workflow together with the data matrix used in the analysis is openly available from Dryad https://doi.org/10.5061/dryad.v41ns1rs5 [81]. Because of local university policy and GDPR, videos and other sensitive material will not be shared.

Authors' contributions. K.A.: drafting the article, conception and design, data analysis and interpretation, final approval of the version to be published. M.L.: data collection, designed the longitudinal study, critical revision of the article, data analysis and interpretation, final approval of the version to be published. L.F.: data collection, critical revision of the article, final approval of the version to be published. B.K.: data collection, critical revision of the article, final approval of the version to be published. M.F.: data collection, critical revision of the article, final approval of the version to be published.

Competing interests. The authors declare that the study was conducted in the absence of any conflicting interests.

Funding. The BASICchild project was funded by the following grants: Gustaf Gredebäck (Wallenberg Fellowship: KAW 2012.0120) and Marcus Lindskog (Riksbankens Jubileumsfond: P15-0430:1).

Acknowledgements. We thank all families that participated and colleagues that have worked with the BASIC project. We also thank everyone who contributed with valuable feedback on the manuscript.

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
