## [Reviewer comments · Royal Society Open Science]

Review History

RSOS-200342.R0 (Original submission)

Review form: Reviewer 1

Is the manuscript scientifically sound in its present form?

Yes

Are the interpretations and conclusions justified by the results?

No

Is the language acceptable?

Yes

Do you have any ethical concerns with this paper?

No

Have you any concerns about statistical analyses in this paper?

No

Recommendation?

Major revision is needed (please make suggestions in comments)

Comments to the Author(s)

Overall I think this paper asks an interesting question, the methods are adequate for answering the question, and the paper is generally written in a clear way. I applaud the transparent reporting with regard to the description of the internal pre-registration process and the timeline of inclusion of variables. I think it is suitable for publication pending the below major revisions mainly relating to the interpretation of the results.

Confidence in results: Overall, I am not very confident in the results that were reported in the paper. Main conclusions are drawn from “marginally significant” and very close to 0.05 p values. Please take out all mention of results as “marginally significant”; if the standard $p < 0.05$ criterion for significance is being used, then results larger than this should not be called marginally significant as this is subjective, and these results should not be interpreted as significant in the discussion & conclusion. This includes the relationship between attachment and gaze following at 10 months, maternal depression at 6 weeks and gaze following at 10 months, maternal depression at 6 months and gaze following at 10 months). In the path model, update this so that there is not a separate way of visualising these results, and simply have paths that are significant vs. paths that aren't. Similarly, interpret with caution results that are very close to the 0.05 threshold (gaze following at 6 and 10 months, attachment and gaze following at 6 months). The strongest conclusions should be drawn from the remaining results that are not close to this threshold: longitudinal relationships between all measures of maternal depression, and the relationship between maternal depression at 12 months and gaze following at 10 months). Please note, this should not be considered to reduce the quality of the paper, as there is no reason that null results should not be reported as they contribute to the literature on this topic. The authors might consider including Bayes Factor analyses for all results in order to be able to make conclusions about the null results and to see whether this can shed some light on the results close on either side of $p=0.05$.

Interpretation of results: As well as changing the strength of the interpretation of the results to be in line with the p values as discussed above, I also think the strength of the claim for a social-first account over a reinforcement theory of gaze following isn't warranted by the results in the paper. It is even stated (correctly) that the current study cannot disentangle the two experience-dependent perspectives, so it is unwarranted to come down on one side of the fence, especially in the title of the paper itself. It is informative to discuss how these results combined with the two papers cited suggest a social-first theory, however this should not be the conclusion of the paper because that is not what is shown by the results in this paper alone. What this paper itself contributes most strongly is the finding of a relationship between maternal depression at 12 months and gaze following at 10 months, and that this supports the experience-dependent perspective.

Data availability: Although the eye-tracking workflow together with the data matrix used in the analysis is openly available, I was unable to open the “BASIC_GF.study” file (I'm unsure of what software is necessary to open this), and the “BASIC_GF.csv” doesn't seem to come with any document explaining the names of any variables or how they have been coded. Additionally, in the body of the paper it says that the analysis script is in the supplementary materials, however I could not find this, as it was not in the OSF project and the alternative DOI led to an error page. Please update the OSF project wiki to include information about software needed to open all files, to explain the content of files as outlined above, and please upload the analysis script to the OSF project.

Review form: Reviewer 2 (Scott Johnson)

Is the manuscript scientifically sound in its present form?

Yes

Are the interpretations and conclusions justified by the results?

Yes

Is the language acceptable?

Yes

Do you have any ethical concerns with this paper?

No

Have you any concerns about statistical analyses in this paper?

No

Recommendation?

Major revision is needed (please make suggestions in comments)

Comments to the Author(s)

This interesting manuscript describes a longitudinal study with infants who were assessed for gaze following at 6 and 10 months and for general attachment status. Measures of postpartum maternal depression were taken at three time points. Infants' gaze following was affected at both ages but for different reasons (attachment status at 6 months and maternal depression at 10 months). Results were interpreted within a theoretical framework that distinguishes between gaze following as an innate ability (and thus impervious to environmental conditions) vs. gaze following as a skill that is shaped by experience with the social world. I thought the paper was well written and clear, and the results are important.

Major comments:

1. I think the theoretical distinction as drawn by the authors is a straw person. Even the most ardent nativist would likely allow for perturbations in development of innate abilities if the environment does not provide the "expected" inputs, in the present case an attentive, consistent caregiver. Examples abound (e.g., institutional care has a negative impact on later social function).
2. In any case I think it has already been shown that development of gaze following in infancy is influenced by different environments and experiences, including research that is cited in the present paper. I am not sure about one of the papers cited, though I might be missing something. In the present ms the authors note that the Senju et al. (2015) study "does not find any indication that gaze following is driven by the amount of experience infants have had with seeing adults," but I don't think that is correct. Rather, results were mixed with respect to comparability of gaze following development in infants of blind parents vs. controls (it was reduced for at least one measure).
3. It is interesting that attachment and maternal depression were unrelated yet each had an influence on gaze following. This might warrant more discussion. On the other hand, I couldn't really follow the argument that learning is not important in gaze following.

Minor comments:

p. 4, line 11: The phrase "first year after birth" is to be preferred to "first year of life," because the latter term discounts prenatal developmental processes. These are particularly important when considering infant development.

p. 12, line 20: "reviled" → "revealed"

Scott P. Johnson, UCLA (I sign all reviews)

Decision letter (RSOS-200342.R0)

Dear Mr Astor:

Manuscript ID RSOS-200342 entitled "Social and emotional contexts predict the development of gaze following in early infancy: A social-first account" which you submitted to Royal Society Open Science, has been reviewed. The comments from reviewers are included at the bottom of this letter.

In view of the criticisms of the reviewers, the manuscript has been rejected in its current form. However, a new manuscript may be submitted which takes into consideration these comments.

Please note that resubmitting your manuscript does not guarantee eventual acceptance, and that your resubmission will be subject to peer review before a decision is made.

Your resubmitted manuscript should be submitted by 11-Dec-2020. If you are unable to submit by this date please contact the Editorial Office.

on behalf of Dr Teodora Gliga (Associate Editor)
openscience@royalsociety.org

Associate Editor Comments to Author (Dr Teodora Gliga):

Associate Editor: 1

Comments to the Author:

The paper was now seen by two researchers highly familiar with the field and I have read it carefully myself. I believe the paper does not yet provide evidence for social context affecting gaze following - which is your main claim. On one hand, you only find non-significant associations between depression and GF and, on the other, you do not provide enough background information in support of attachment styles necessarily reflecting differences in parental input. Many of our readers will not be familiar with the literature on attachment and may wonder whether differences intrinsic to the child could drive both attachment styles and differences in GF. There was very little discussion of which attachment style is beneficial and

why. A stronger motivation for looking at attachment to investigate experience dependent processes, in particular the social-first hypothesis, needs to be given.

If you decide to resubmit the paper, please also revise in agreement with reviewer comments, paying particular attention to

- 1) your interpretation of non-significant findings - they need to be discussed as absence of evidence. To help readers judge of your effect sizes (for significant and non-significant associations), please report beta values together with their confidence intervals
- 2) how you review literature - in particular with respect to reporting Senju et al findings - they do find first looks unaffected but a difference in amount of time spent on the gazed at object; a distinction between the two measures was made in the literature, with the former reported as less affected by environment or by atypical development. Discussing this distinction would strengthen your findings of a potential environmental contribution.

Yu may also want to cite:

Peña, M., Arias, D., & Dehaene-Lambertz, G. (2014). Gaze following is accelerated in healthy preterm infants. *Psychological science*, 25(10), 1884-1892.

I have an additional suggestion which is to use a scaled GF measure (correct-incorrect)/(correct +incorrect) rather than the difference scores. This measure better takes into account individual differences in the amount of valid trials.

Reviewers' Comments to Author:

Reviewer: 1

Comments to the Author(s)

Overall I think this paper asks an interesting question, the methods are adequate for answering the question, and the paper is generally written in a clear way. I applaud the transparent reporting with regard to the description of the internal pre-registration process and the timeline of inclusion of variables. I think it is suitable for publication pending the below major revisions mainly relating to the interpretation of the results.

Confidence in results: Overall, I am not very confident in the results that were reported in the paper. Main conclusions are drawn from "marginally significant" and very close to 0.05 p values. Please take out all mention of results as "marginally significant"; if the standard $p < 0.05$ criterion for significance is being used, then results larger than this should not be called marginally significant as this is subjective, and these results should not be interpreted as significant in the discussion & conclusion. This includes the relationship between attachment and gaze following at 10 months, maternal depression at 6 weeks and gaze following at 10 months, maternal depression at 6 months and gaze following at 10 months). In the path model, update this so that there is not a separate way of visualising these results, and simply have paths that are significant vs. paths that aren't. Similarly, interpret with caution results that are very close to the 0.05 threshold (gaze following at 6 and 10 months, attachment and gaze following at 6 months). The strongest conclusions should be drawn from the remaining results that are not close to this threshold: longitudinal relationships between all measures of maternal depression, and the relationship between maternal depression at 12 months and gaze following at 10 months). Please note, this should not be considered to reduce the quality of the paper, as there is no reason that null results should not be reported as they contribute to the literature on this topic. The authors might consider including Bayes Factor analyses for all results in order to be able to make conclusions about the null results and to see whether this can shed some light on the results close on either side of $p = 0.05$.

Interpretation of results: As well as changing the strength of the interpretation of the results to be in line with the p values as discussed above, I also think the strength of the claim for a social-first account over a reinforcement theory of gaze following isn't warranted by the results in the paper. It is even stated (correctly) that the current study cannot disentangle the two experience-

dependent perspectives, so it is unwarranted to come down on one side of the fence, especially in the title of the paper itself. It is informative to discuss how these results combined with the two papers cited suggest a social-first theory, however this should not be the conclusion of the paper because that is not what is shown by the results in this paper alone. What this paper itself contributes most strongly is the finding of a relationship between maternal depression at 12 months and gaze following at 10 months, and that this supports the experience-dependent perspective.

Data availability: Although the eye-tracking workflow together with the data matrix used in the analysis is openly available, I was unable to open the "BASIC_GF.study" file (I'm unsure of what software is necessary to open this), and the "BASIC_GF.csv" doesn't seem to come with any document explaining the names of any variables or how they have been coded. Additionally, in the body of the paper it says that the analysis script is in the supplementary materials, however I could not find this, as it was not in the OSF project and the alternative DOI led to an error page. Please update the OSF project wiki to include information about software needed to open all files, to explain the content of files as outlined above, and please upload the analysis script to the OSF project.

Reviewer: 2

Comments to the Author(s)

This interesting manuscript describes a longitudinal study with infants who were assessed for gaze following at 6 and 10 months and for general attachment status. Measures of postpartum maternal depression were taken at three time points. Infants' gaze following was affected at both ages but for different reasons (attachment status at 6 months and maternal depression at 10 months). Results were interpreted within a theoretical framework that distinguishes between gaze following as an innate ability (and thus impervious to environmental conditions) vs. gaze following as a skill that is shaped by experience with the social world. I thought the paper was well written and clear, and the results are important.

Major comments:

1. I think the theoretical distinction as drawn by the authors is a straw person. Even the most ardent nativist would likely allow for perturbations in development of innate abilities if the environment does not provide the "expected" inputs, in the present case an attentive, consistent caregiver. Examples abound (e.g., institutional care has a negative impact on later social function).
2. In any case I think it has already been shown that development of gaze following in infancy is influenced by different environments and experiences, including research that is cited in the present paper. I am not sure about one of the papers cited, though I might be missing something. In the present ms the authors note that the Senju et al. (2015) study "does not find any indication that gaze following is driven by the amount of experience infants have had with seeing adults," but I don't think that is correct. Rather, results were mixed with respect to comparability of gaze following development in infants of blind parents vs. controls (it was reduced for at least one measure).
3. It is interesting that attachment and maternal depression were unrelated yet each had an influence on gaze following. This might warrant more discussion. On the other hand, I couldn't really follow the argument that learning is not important in gaze following.

Minor comments:

- p. 4, line 11: The phrase "first year after birth" is to be preferred to "first year of life," because the latter term discounts prenatal developmental processes. These are particularly important when considering infant development.

p. 12, line 20: "reviled" → "revealed"

Scott P. Johnson, UCLA (I sign all reviews)

Author's Response to Decision Letter for (RSOS-200342.R0)

See Appendix A.

RSOS-201178.R0

Review form: Reviewer 1

Is the manuscript scientifically sound in its present form?

Yes

Are the interpretations and conclusions justified by the results?

Yes

Is the language acceptable?

Yes

Do you have any ethical concerns with this paper?

No

Have you any concerns about statistical analyses in this paper?

No

Recommendation?

Accept with minor revision (please list in comments)

Comments to the Author(s)

I am happy with the revision of the manuscript and feel that it makes an important contribution to the literature. Just a tiny thing, you may wish to also cite this recent proceedings paper in the introduction when talking about the reinforcement learning theory of gaze following: Silverstein, Westermann, Parise & Twomey, 2019

Review form: Reviewer 2 (Scott Johnson)

Is the manuscript scientifically sound in its present form?

Yes

Are the interpretations and conclusions justified by the results?

Yes

Is the language acceptable?

Yes

Do you have any ethical concerns with this paper?

No

Have you any concerns about statistical analyses in this paper?

No

Recommendation?

Accept with minor revision (please list in comments)

Comments to the Author(s)

I thought the revised manuscript was much improved. Just a few comments.

p. 14, footnote 1: "is" → "are" (two places)

p. 15, line 21: "of life" → "after birth"

p. 16: The attempt to elucidate findings of Senju et al. and Hernik & Broesch is appreciated, but I found this section hard to follow. It might be helpful to describe infants' behaviors in the two studies in more detail. For example, a reader might be confused to read about following the gaze of blind individuals.

p. 18, lines 17-19: I couldn't really follow this.

Scott P. Johnson, UCLA (I sign all reviews)

Decision letter (RSOS-201178.R0)

Dear Mr Astor,

On behalf of the Editor, I am pleased to inform you that your Manuscript RSOS-201178 entitled "Social and emotional contexts predict the development of gaze following in early infancy: A social-first account" has been accepted for publication in Royal Society Open Science subject to minor revision in accordance with the referee suggestions. Please find the referees' comments at the end of this email.

The reviewers and Subject Editor have recommended publication, but also suggest some minor revisions to your manuscript. Therefore, I invite you to respond to the comments and revise your manuscript.

- Ethics statement

- Data accessibility

It is a condition of publication that all supporting data are made available either as supplementary information or preferably in a suitable permanent repository. The data

accessibility section should state where the article's supporting data can be accessed. This section should also include details, where possible of where to access other relevant research materials such as statistical tools, protocols, software etc can be accessed. If the data has been deposited in an external repository this section should list the database, accession number and link to the DOI for all data from the article that has been made publicly available. Data sets that have been deposited in an external repository and have a DOI should also be appropriately cited in the manuscript and included in the reference list.

If you wish to submit your supporting data or code to Dryad (<http://datadryad.org/>), or modify your current submission to dryad, please use the following link:
<http://datadryad.org/submit?journalID=RSOS&manu=RSOS-201178>

- **Competing interests**

- **Authors' contributions**

- **Acknowledgements**

- **Funding statement**

Because the schedule for publication is very tight, it is a condition of publication that you submit the revised version of your manuscript before 06-Aug-2020. Please note that the revision deadline will expire at 00.00am on this date. If you do not think you will be able to meet this date please let me know immediately.

on behalf of Dr Teodora Gliga (Associate Editor)
openscience@royalsociety.org

Associate Editor Comments to Author (Dr Teodora Gliga):

I have now received comments back from the two original reviewers, who were satisfied with your revised manuscript. I also appreciated the additional information provided on the mechanisms through which attachment may work and the careful consideration of the various ways in which gaze following can be computed.

I therefore accept your manuscript for publication, pending the additional clarifications requested by Reviewer 2, in particular with respect to clarifying how Senju and Herrman support a social first hypothesis. While I remain unconvinced by this account, this is not problematic as this is only a secondary aim of this manuscript, which clearly brings support for experience affecting GF ability during development.

Alternatively, you may want to look at the number of valid trials. If these are decided, as in previous work, based on infants not attending to the experimenter (see below for having to provide this missing information), they may indicate social motivation; you'd therefore expect depression/attachment to also predict the number of valid trials. In the absence of this evidence, simply indicate that support for a motivational account remains indirect and remove "a social-first account" from your title (something one of the reviewers had requested in the first round of reviews).

Please also address the few additional comments below:

1. indicate how trial validity was determined
2. parents in Senju study were blind, not deaf (p. 16)
3. might there be a better way of referring to families than high-functioning (that seems to set a high bar for parenting); well-functioning ?

Finally, just a comment on your interpretation of first look as the primary index of an ability to follow gaze. While I agree that the direction of first looks indexes the ability to read gaze direction, given one of the functions of gaze following is for learning about the world, longer looking to gazed at objects, following the gaze shifts, seems to be the key outcome of this process, and is indeed a better predictor of later abilities than first look direction itself (for example, Parsons et al, 2019).

Reviewer comments to Author:

Reviewer: 2

Comments to the Author(s)

I thought the revised manuscript was much improved. Just a few comments.

p. 14, footnote 1: "is" → "are" (two places)

p. 15, line 21: "of life" → "after birth"

p. 16: The attempt to elucidate findings of Senju et al. and Hernik & Broesch is appreciated, but I found this section hard to follow. It might be helpful to describe infants' behaviors in the two studies in more detail. For example, a reader might be confused to read about following the gaze of blind individuals.

p. 18, lines 17-19: I couldn't really follow this.

Scott P. Johnson, UCLA (I sign all reviews)

Reviewer: 1

Comments to the Author(s)

I am happy with the revision of the manuscript and feel that it makes an important contribution to the literature. Just a tiny thing, you may wish to also cite this recent proceedings paper in the introduction when talking about the reinforcement learning theory of gaze following: Silverstein, Westermann, Parise & Twomey, 2019

Author's Response to Decision Letter for (RSOS-201178.R0)

See Appendix B.

Decision letter (RSOS-201178.R1)

Dear Mr Astor,

It is a pleasure to accept your manuscript entitled "Social and emotional contexts predict the development of gaze following in early infancy" in its current form for publication in Royal Society Open Science.

on behalf of Dr Teodora Gliga (Associate Editor) and Andrew Dunn (Subject Editor)
openscience@royalsociety.org

Appendix A

ANSWER: Before responding to the comments, we would like to thank you for inviting us to resubmit our manuscript.

Associate Editor Comments to Author (Dr Teodora Gliga):

Associate Editor: 1

Comments to the Author:

The paper was now seen by two researchers highly familiar with the field and I have read it carefully myself. I believe the paper does not yet provide evidence for social context affecting gaze following – which is your main claim. On one hand, you only find non-significant associations between depression and GF and, on the other, you do not provide enough background information in support of attachment styles necessarily reflecting differences in parental input. Many of our readers will not be familiar with the literature on attachment and may wonder whether differences intrinsic to the child could drive both attachment styles and differences in GF. There was very little discussion of which attachment style is beneficial and why. A stronger motivation for looking at attachment to investigate experience dependent processes, in particular the social-first hypothesis, needs to be given.

ANSWER: Thank you for the insight on information on attachment. This is important and has now been clarified when attachment is first introduced in the introduction (that attachment style is thought to be primarily driven by mothers' response to their infant). Regarding which attachment style that is beneficial and why, this is not our focus (as long as it is clear that a secure attachment is qualitatively different from a non-secure attachment). In this study we are interested in attachment as a measure of the infants' social environment. We have tried to make this clearer in the current version.

Based on your comment, "*On one hand, you only find non-significant associations between depression and GF ...*" we realize that this is a complex model using a lot of measures of development and it is easy that parts get lost. We would, however, like to emphasize that there is a significant association between gaze following at 10 months and depression, both in the zero-order correlation and the final model ($r_s = -.33, p < .01$). We see this as one of the strengths of this study and one of the core reasons for claiming that our social first account has some validity. We have tried our best to enhance the clarity of argumentation in order to make this result is more accessible.

If you decide to resubmit the paper, please also revise in agreement with reviewer comments, paying particular attention to

1) your interpretation of non-significant findings - they need to be discussed as absence of evidence. To help readers judge of your effect sizes (for significant and non-significant associations), please report beta values together with their confidence intervals

ANSWER: Non-significant findings are no longer mentioned as 'marginally significant'. Standard error has been added to Table S2 and the figure has been updated and now contain p-values (instead of *indications) to accompany the path coefficients.

2) how you review literature - in particular with respect to reporting Senju et al findings - they do find first looks unaffected but a difference in amount of time spent on the gazed at object; a distinction between the two measures was made in the literature, with the former reported as less affected by

environment or by atypical development. Discussing this distinction would strengthen your findings of a potential environmental contribution.

ANSWER: We have used the ‘first gaze shift’ because it is the most often used measure of gaze following in the literature, and also the core ability that we are interested in (the degree to which infants follow others gaze). We have added a brief discussion on alternative measures to the Methods section together with an elaboration on the different methods of score aggregation. In addition, we must confess that we are not sure how to interpret looking time measures in this context. It is always possible to add or discuss more measures, but this also means that the length of the paper, and the complexity of the findings, increase substantially without adding significantly to the core question; is the tendency to follow gaze experience dependent and what are the principles that govern this ability?

As a side note we would also like to add a reference to the paper that you refer to above (Senju et al., 2015). In this paper it is stated that *“infants of blind parents allocated less attention to adult eye movements and gaze direction, an effect that increased between 6–10 and 12–16 months of age.”* However, there is no significant effect at 6-10 months, only at 12-16 (p. 3088 first paragraph), which is outside of the age range we target in this study and outside the window of early developing gaze following. Later however, they discuss their findings as *“indicating that atypical experience of gaze communication does not have a major impact on initial eye gaze processing during the first year of life but rather has increasing developmental impact beyond the first birthday.”* (p. 3088 first paragraph). We are highly impressed by this study, and refer to it heavily in our manuscript, but it is not clear to us how this measure should be interpreted or if it is indeed the most suitable measure to assess gaze following.

Yu may also want to cite:

Peña, M., Arias, D., & Dehaene-Lambertz, G. (2014). Gaze following is accelerated in healthy preterm infants. *Psychological science*, 25(10), 1884-1892.

ANSWER: Thanks for suggesting this paper. It is clearly an important and highly valuable one that has been added to the introduction.

I have an additional suggestion which is to use a scaled GF measure (correct-incorrect)/(correct +incorrect) rather than the difference scores. This measure better takes into account individual differences in the amount of valid trials.

ANSWER: Thank you for the suggestion. We reran the analyses using the proportion score you suggest, in accordance with e.g. Senju and Csibra. This approach yielded the same results as those reported in the original manuscript. We now report on both measures in the paper. Our original model (based on the difference score) in the main text and the alternative model (based on the proportion score) in a footnote.

We believe that both measures have its strengths and weaknesses and that none of the measures captures individual differences better than the other does but they tend to converge (this is also what the results from the alternative model, based on the proportion score, suggest).

The reason why we choose to emphasize the difference score (traditionally most common) is related to this being part of a larger longitudinal project: When using a difference score, a less strict inclusion criterion can be used in order to prioritize the integrity of the data matrix,

diminishing the number of missing data points. This is a strength in field research and longitudinal work where data collection generally offers more challenges.

To elaborate on this, the risk with a proportion score is that less reliable data (participants who contributed few data points) will introduce more noise. A participant with only a few trials of data will be more likely to receive a 100% correct gaze following with a proportion score, whereas participants with few trials will be weighted less when using difference scores (1 trials correct, 0 trials incorrect = 100% correct with a proportion score but a difference score of 1 in a range from -4 to 4 with difference score). In classical experiments participants with few trials can be removed and replaced, but in longitudinal work this is less of an option. In this case we have chosen to focus on a method that includes as many participants as possible rather than using a relative score with a tough inclusion criteria (which is what was done by Senju et al).

Given that there is no critical reason to change our methodology, we wish to be consistent with earlier publications from our lab, and with other studies in the doctoral theses that this study will be part of, and with the longitudinal project that the data is drawn from.

An additional (minor) matter that we would like to bring to your attention.

Before continuing to answer reviewer comments, it should be mentioned that we discovered an error in the methods section that have now been addressed. The inclusion criterion was described as being two trials when in fact it is one trial. We also elaborated the choice of criteria as it depends on which measure on gaze following one use (difference/proportion-score).

Reviewers' Comments to Author:

Reviewer: 1

Comments to the Author(s)

Overall I think this paper asks an interesting question, the methods are adequate for answering the question, and the paper is generally written in a clear way. I applaud the transparent reporting with regard to the description of the internal pre-registration process and the timeline of inclusion of variables. I think it is suitable for publication pending the below major revisions mainly relating to the interpretation of the results.

Confidence in results: Overall, I am not very confident in the results that were reported in the paper. Main conclusions are drawn from "marginally significant" and very close to 0.05 p values. Please take out all mention of results as "marginally significant"; if the standard $p < 0.05$ criterion for significance is being used, then results larger than this should not be called marginally significant as this is subjective, and these results should not be interpreted as significant in the discussion & conclusion. This includes the relationship between attachment and gaze following at 10 months, maternal depression at 6 weeks and gaze following at 10 months, maternal depression at 6 months and gaze following at 10 months).

ANSWER: Thank you for this comment, we agree that non-significant results should be reported as such. However, we want to point out that the main conclusions are not drawn from "marginally significant" results. We only discussed the significant path from depression 12 -> gaze following 10

and Attachment -> GF 6 as support of the hypothesis. Nevertheless, we agree that statements as “marginally significant” is better removed such that readers can judge for themselves. This has now been addressed.

In the path model, update this so that there is not a separate way of visualising these results, and simply have paths that are significant vs. paths that aren't.

ANSWER: This is changed as you suggested and the exact p-values have now replaced the *indicators from the previous model.

Similarly, interpret with caution results that are very close to the 0.05 threshold (gaze following at 6 and 10 months, attachment and gaze following at 6 months). The strongest conclusions should be drawn from the remaining results that are not close to this threshold: longitudinal relationships between all measures of maternal depression, and the relationship between maternal depression at 12 months and gaze following at 10 months).

ANSWER: Following the logic of a strict criteria of $p < .05$ for significance, disregarding anything not reaching the threshold as non-significant (as you reasonably suggested), also suggest that we should accept values that do pass that threshold.

Please note, this should not be considered to reduce the quality of the paper, as there is no reason that null results should not be reported as they contribute to the literature on this topic. The authors might consider including Bayes Factor analyses for all results in order to be able to make conclusions about the null results and to see whether this can shed some light on the results close on either side of $p=0.05$.

ANSWER: Thank you for the many good suggestions on how to approach the data. Though we agree that Bayes Factor can be very useful in some cases, we are not convinced that introducing it here is likely to help us understand the results. Bayes factor is generally quite consistent with frequentist statistics and it is not likely that p-values between .05 and .1 will support the null hypothesis using this method. We have already added one additional model to the new manuscript (described in a footnote and discussed in the methods section). Adding even more models/measures could make the article very voluminous and perhaps less accessible. In agreement with your first suggestion, we have taken note on applying a more stringent approach to the p-values.

Interpretation of results: As well as changing the strength of the interpretation of the results to be in line with the p values as discussed above, I also think the strength of the claim for a social-first account over a reinforcement theory of gaze following isn't warranted by the results in the paper. It is even stated (correctly) that the current study cannot disentangle the two experience-dependent perspectives, so it is unwarranted to come down on one side of the fence, especially in the title of the paper itself. It is informative to discuss how these results combined with the two papers cited suggest a social-first theory, however this should not be the conclusion of the paper because that is not what is shown by the results in this paper alone. What this paper itself contributes most strongly is the finding of a relationship between maternal depression at 12 months and gaze following at 10 months, and that this supports the experience-dependent perspective.

ANSWER: This is a valid comment and we have now toned down our interpretation, making it come through as less forceful. However, though we state that the current study cannot disentangle the two experience-dependent perspectives, we do think it converges with previous research to make a case against the reinforcement theory of gaze following and at the same time being

consistent with a social-first account. We have elaborated on this argument in the discussion as we realized it was lacking. Nevertheless, we see your point and agree that we should be more cautious making ours.

Data availability: Although the eye-tracking workflow together with the data matrix used in the analysis is openly available, I was unable to open the "BASIC_GF.study" file (I'm unsure of what software is necessary to open this), and the "BASIC_GF.csv" doesn't seem to come with any document explaining the names of any variables or how they have been coded. Additionally, in the body of the paper it says that the analysis script is in the supplementary materials, however I could not find this, as it was not in the OSF project and the alternative DOI led to an error page. Please update the OSF project wiki to include information about software needed to open all files, to explain the content of files as outlined above, and please upload the analysis script to the OSF project.

ANSWER: Unfortunately, the uwid does not seem to work properly and has been removed. In the methods section under "Gaze following data analysis" it is described that we use TimeStudio under MATLAB to analyze the eye-tracking data, thus you will need those programs to run the analysis. Note that TimeStudio is open source and free and runs under MATLAB. The supplementary materials contain the scripts and coding descriptions that you asked for. It is unfortunate that you did not get the chance to read it.

Thank you for your valuable comments! Though we may not fully agree on all points, your comments helped us improve the manuscript substantially and we hope that you will be satisfied with the result.

Reviewer: 2

Comments to the Author(s)

This interesting manuscript describes a longitudinal study with infants who were assessed for gaze following at 6 and 10 months and for general attachment status. Measures of postpartum maternal depression were taken at three time points. Infants' gaze following was affected at both ages but for different reasons (attachment status at 6 months and maternal depression at 10 months). Results were interpreted within a theoretical framework that distinguishes between gaze following as an innate ability (and thus impervious to environmental conditions) vs. gaze following as a skill that is shaped by experience with the social world. I thought the paper was well written and clear, and the results are important.

Major comments:

1. I think the theoretical distinction as drawn by the authors is a straw person. Even the most ardent nativist would likely allow for perturbations in development of innate abilities if the environment does not provide the "expected" inputs, in the present case an attentive, consistent caregiver. Examples abound (e.g., institutional care has a negative impact on later social function).

ANSWER: We see your point, but would like to add a different perspective. It is not at all obvious that gaze following is affected by the environment. In fact, many would argue that it is not, and given that gaze following is an ability that can be detected in many animal species suggest that it might (at least in early development) be a fundamental pre-potent skill. Moreover, given that gaze following is such an important ability, it is crucial to understand both if and how everyday experiences, such as depression and a lack of connection, can affect this ability. In the current version of the manuscript we have attempted to make our perspective clearer. We hope that the current version will convince reviewers that the question, is at least worth asking .

2. In any case I think it has already been shown that development of gaze following in infancy is influenced by different environments and experiences, including research that is cited in the present paper. I am not sure about one of the papers cited, though I might be missing something. In the present ms the authors note that the Senju et al. (2015) study “does not find any indication that gaze following is driven by the amount of experience infants have had with seeing adults,” but I don’t think that is correct. Rather, results were mixed with respect to comparability of gaze following development in infants of blind parents vs. controls (it was reduced for at least one measure).

ANSWER: Thank you for this comment, it is clear that we need to elaborate on this point and have tried to do so in the manuscript. Senju et al (2015) did not find a difference in the principal measure of gaze following, the differential looking score. They did find a difference in looking time to the objects, but only in infants older than 1 year. This is unfortunately not very clear in that paper.

3. It is interesting that attachment and maternal depression were unrelated yet each had an influence on gaze following. This might warrant more discussion. On the other hand, I couldn’t really follow the argument that learning is not important in gaze following.

ANSWER: We have now included a section discussing the lack of relation between maternal depression and attachment in the discussion.

We also agree that the argument against a reinforcement process was lacking. We have elaborated on this and tried to make this argument much clearer.

Minor comments:

p. 4, line 11: The phrase “first year after birth” is to be preferred to “first year of life,” because the latter term discounts prenatal developmental processes. These are particularly important when considering infant development.

ANSWER: Changed

p. 12, line 20: “reviled” —> “revealed”

ANSWER: Changed

Scott P. Johnson, UCLA (I sign all reviews)

Thank you for inviting us to clarify on a few important matters that came out as unclear/unanswered in the original manuscript. We hope that you will be satisfied with the revised version.

Appendix B

Dear Dr Teodora Gliga,

Thank you for accepting our manuscript for publication in Royal Society Open Science. The manuscript has been revised and clarified in accordance with your requests, and uploaded in two versions, one clean version and one with tracked changes.

Best regards, Kim